# Diffusion Models for Constrained Domains

**Nic Fishman**                                                          *njwfish@gmail.com*
*Department of Statistics, University of Oxford*

**Leo Klarner**                                                    *leo.klarner@stats.ox.ac.uk*
*Department of Statistics, University of Oxford*

**Valentin De Bortoli**                                          *valentin.debortoli@gmail.com*
*CNRS, ENS Ulm*

**Emile Mathieu**                                                        *ebm32@cam.ac.uk*
*Department of Engineering, University of Cambridge*

**Michael Hutchinson**                                    *michael.hutchinson@stats.ox.ac.uk*
*Department of Statistics, University of Oxford*

**Reviewed on OpenReview:** *https://openreview.net/forum?id=xuWTFQ4VG0*

## Abstract

Denoising diffusion models are a novel class of generative algorithms that achieve state-of-the-art performance across a range of domains, including image generation and text-to-image tasks. Building on this success, diffusion models have recently been extended to the Riemannian manifold setting, broadening their applicability to a range of problems from the natural and engineering sciences. However, these Riemannian diffusion models are built on the assumption that their forward and backward processes are well-defined for all times, preventing them from being applied to an important set of tasks that consider manifolds defined via a set of inequality constraints. In this work, we introduce a principled framework to bridge this gap. We present two distinct noising processes based on (i) the *logarithmic barrier* metric and (ii) the *reflected* Brownian motion induced by the constraints. As existing diffusion model techniques cannot be applied in this setting, we derive new tools to define such models in our framework. We then demonstrate the practical utility of our methods on a number of synthetic and real-world tasks, including applications from robotics and protein design.

## 1 Introduction

Diffusion models (Sohl-Dickstein et al., 2015; Song & Ermon, 2019; Song et al., 2021; Ho et al., 2020) have recently been introduced as a powerful new paradigm for generative modelling. They work as follows: noise is progressively added to data following a Stochastic Differential Equation (SDE)—the forward *noising* process—until it is approximately Gaussian. The generative model is given by an approximation of the associated *time-reversed* process called the backward *denoising* process. This is also an SDE whose drift depends on the gradient of the logarithmic densities of the forward process, referred to as the Stein score. This score is approximated by leveraging techniques from deep learning and score matching (Hyvärinen, 2005; Vincent, 2011). Building on the success of diffusion models in domains such as images and text, this framework has recently been extended to a wide range of Riemannian manifolds (De Bortoli et al., 2022; Huang et al., 2022), broadening their applicability to various important modelling domains from the natural and engineering sciences—including Lie groups such as the group of rotations SO(3), the group of rigid body motions SE(3), and many others (see e.g. Trippe et al., 2022; Corso et al., 2022; Watson et al., 2022; Leach et al., 2022; Urain et al., 2022; Yim et al., 2023).

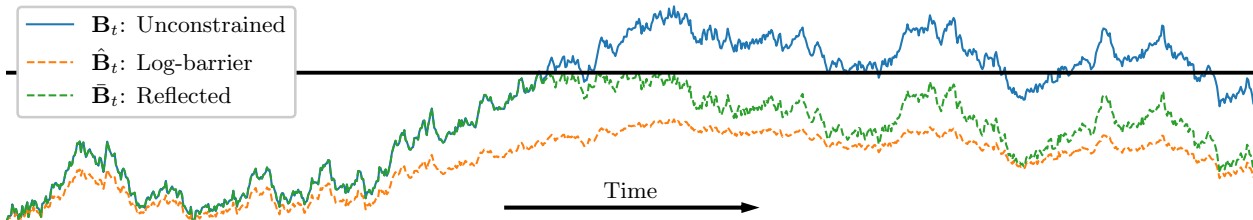

Figure 1: The behaviour of different types of noising processes considered in this work defined on the unit interval. $\mathbf{B}_t$: Unconstrained (Euclidean) Brownian motion. $\hat{\mathbf{B}}_t$: Log-barrier forward noising process. $\bar{\mathbf{B}}_t$: Reflected Brownian motion. All sampled with the same initial point and driving noise. Black line indicates the boundary.

However, a key assumption of the Riemannian diffusion models introduced in De Bortoli et al. (2022) and Huang et al. (2022) is that the stochastic processes they consider are defined *for all times*. While this holds for a large class of stochastic processes, it is not the case for most manifolds defined via a set of inequality constraints. For instance, in the case of the hypercube $(-1,1)^d$ equipped with the Euclidean metric, the Riemannian Brownian motion coincides with the Euclidean $d$-dimensional Brownian motion $(\mathbf{B}_t)_{t \in [0,T]}$ as long as $\mathbf{B}_t \in (-1, 1)$. With probability one, $(\mathbf{B}_t)_{t \geq 0}$ escapes from $(-1, 1)$, meaning that the Riemannian Brownian motion is not defined for all times and the frameworks introduced in De Bortoli et al. (2022) and Huang et al. (2022) do not apply. Such constrained manifolds comprise a wide variety of settings—including polytopes and convex sets of Euclidean spaces—and are studied across a large number of disciplines, ranging from computational statistics (Morris, 2002), over robotics (Han & Rudolph, 2006) and quantum physics (Lukens et al., 2020), to computational biology (Thiele et al., 2013). Deriving principled diffusion models that are able to operate directly on these manifolds is thus of significant practical importance, as they enable generative modelling in data-scarce and safety-critical settings in which constraints on the modelled domain may reduce the number of degrees of freedom or prevent unwanted behaviour.

As sampling problems on such manifolds are important (Kook et al., 2022; Heirendt et al., 2019), a flurry of Markov chain based methods have been developed to sample from unnormalised densities. Successful algorithms include the reflected Brownian motion (Williams, 1987; Petit, 1997; Shkolnikov & Karatzas, 2013), log-barrier methods (Kannan & Narayanan, 2009; Lee & Vempala, 2017; Noble et al., 2022; Kook et al., 2022; Gatmiry & Vempala, 2022; Lee & Vempala, 2018) and hit-and-run approaches in the case of polytopes (Smith, 1984; Lovász & Vempala, 2006). In this work, we study the generative modelling counterparts of these algorithms through the lens of diffusion models. Among existing methods for statistical sampling on constrained manifolds, the geodesic Brownian motion (Lee & Vempala, 2017) and the reflected Brownian motion (Williams, 1987) are continuous stochastic processes, and thus well suited for extending the continuous Riemannian diffusion framework developed by De Bortoli et al. (2022) and Huang et al. (2022). In particular, we introduce two principled diffusion models for generative modelling on constrained domains based on (i) the geodesic Brownian motion, leveraging tools from the log-barrier methods, and (ii) the reflected Brownian motion. In both cases, we show how one can extend the ideas of time-reversal and score matching to these settings. We demonstrate the practical utility of these methods on a range of tasks defined on convex polytopes and the space of symmetric positive definite matrices, including the constrained conformational modelling of proteins and robotic arms. The code for all of our experiments is available here.

## 2 Background

**Riemannian manifolds.** A Riemannian manifold is a tuple $(\mathcal{M}, \mathfrak{g})$ with $\mathcal{M}$ a smooth manifold and $\mathfrak{g}$ a metric which defines an inner product on tangent spaces. The metric $\mathfrak{g}$ induces key quantities on the manifold, such as an exponential map $\exp_x : \mathrm{T}_x \mathcal{M} \to \mathcal{M}$, defining the notion of following straight lines on manifolds, a gradient operator $\nabla$[1] and a divergence operator $\mathrm{div}$[2]. It also induces the Laplace-Beltrami operator $\Delta$ and

---

[1] The (Riemannian) gradient $\nabla$ is defined s.t. for any smooth $f \in \mathrm{C}^\infty(\mathcal{M})$, $x \in \mathcal{M}, v \in \mathrm{T}_x \mathcal{M}$, $\mathfrak{g}(x)(f(x), v) = \mathrm{d}f(x)(v)$.
[2] The Riemannian divergence acts vector fields and can be defined using the volume form of $\mathcal{M}$.

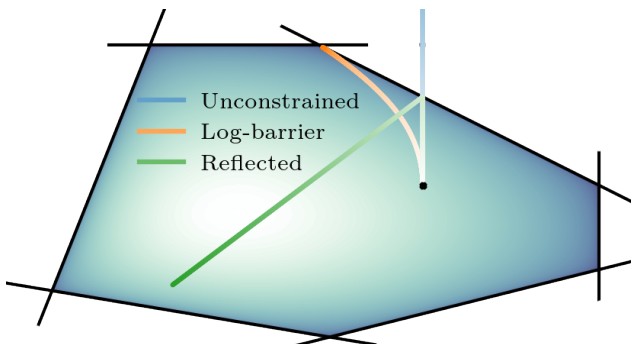

Figure 2: A convex polytope defined by six constraints $\{f_i\}_{i \in \mathcal{I}}$, along with the log barrier potential, and 'straight trajectories' under the log-barrier metric and under the Euclidean metric with and without reflection at the boundary.

consequently a Brownian motion with density (w.r.t. the volume form[3]), whose density is given by the heat equation $\partial_t p_t = \Delta p_t$. We refer the reader to Appendix A for a brief introduction to differential geometry, to Lee (2013) for a thorough treatment and to Hsu (2002) for details on stochastic analysis on manifolds.

**Constrained manifolds.** In this work, we are concerned with *constrained* manifolds. More precisely, given a Riemannian manifold $(\mathcal{N}, \mathfrak{h})$, we consider a family of real functions $\{f_i : \mathcal{N} \to \mathbb{R}\}_{i \in \mathcal{I}}$ indexed by $\mathcal{I}$. We then define

$$\mathcal{M} = \{x \in \mathcal{N} \: : \: f_i(x) < 0, \; i \in \mathcal{I}\}. \tag{1}$$

In this scenario, $\{f_i\}_{i \in \mathcal{I}}$ is interpreted as a set of constraints on $\mathcal{N}$. For example, choosing $\mathcal{N} = \mathbb{R}^d$ and affine constraints $f_i(x) = \langle a_i, x \rangle - b_i$, $x \in \mathbb{R}^d$, we get that $\mathcal{M}$ is an open polytope as illustrated in Figure 2. This setting naturally appears in many areas of engineering, biology, and physics (Boyd et al., 2004; Han & Rudolph, 2006; Lukens et al., 2020).

While the two methods we introduce in Section 3 can be applied to the general framework (1), in our applications, we focus on two specific settings: (a) Polytopes—$\mathcal{N} = \mathbb{R}^d$ and (b) symmetric positive-definite (SPD) matrices under trace constraints—$\mathcal{N} = \mathcal{S}_{++}^d$.

**Continuous diffusion models.** We briefly recall the framework for constructing continuous diffusion processes introduced by Song et al. (2021) in the context of generative modelling over $\mathbb{R}^d$. At minimum diffusion models need four things: (i) A forward noising process converging to an invariant distribution. (ii) A time reversal for the reverse process. (iii) A discretization of the continuous-time process for the forward/reverse process. (iv) A score matching loss — in this paper we will focus on the implicit score matching loss. Song et al. (2021) consider a forward *noising process* $(\mathbf{X}_t)_{t \in [0,T]}$ which progressively noises a data distribution $p_0$ into a Gaussian N(0, Id). More precisely $(\mathbf{X}_t)_{t \in [0,T]}$ is an Ornstein–Uhlenbeck (OU) process which is given by the following stochastic differential equation (SDE)

$$d\mathbf{X}_t = -\tfrac{1}{2}\mathbf{X}_t dt + d\mathbf{B}_t, \qquad \mathbf{X}_0 \sim p_0.$$

Under mild conditions on $p_0$, the time-reversed process $(\overleftarrow{\mathbf{X}}_t)_{t \in [0,T]} = (\mathbf{X}_{T-t})_{t \in [0,T]}$ also satisfies an SDE (Cattiaux et al., 2021; Haussmann & Pardoux, 1986) given by

$$d\overleftarrow{\mathbf{X}}_t = \{\tfrac{1}{2}\overleftarrow{\mathbf{X}}_t + \nabla \log p_{T-t}(\overleftarrow{\mathbf{X}}_t)\}dt + d\mathbf{B}_t, \qquad \overleftarrow{\mathbf{X}}_0 \sim p_T, \tag{2}$$

where $p_t$ denotes the density of $\mathbf{X}_t$. This construction allows direct sampling of the forward process and leverages the Euler-Maruyama discretisation to facilitate sampling of the reverse process. Finally, the quantity $\nabla \log p_t$ is referred as the Stein score and is unavailable in practice. It can be approximated with a score network $s_\theta(t, \cdot)$ trained by minimising a denoising score matching (dsm) loss

$$\mathcal{L}(\theta) = \mathbb{E}[\lambda_t \|\nabla \log p_{t|0}(\mathbf{X}_t | \mathbf{X}_0) - s_\theta(t, \mathbf{X}_t)\|^2], \tag{3}$$

---

[3]We assume that $\mathcal{M}$ is orientable and therefore that a volume form exists.

or an equivalent implicit score matching (ism) loss

$$\mathcal{L}(\theta) = \mathbb{E}[\lambda_t\{\tfrac{1}{2}\|s_\theta(t,\mathbf{X}_t)\|^2 + \text{div}(s_\theta)(t,\mathbf{X}_t)\}] + C, \tag{4}$$

where $C \geq 0$ and $\lambda_t > 0$ is a weighting function, and the expectation is taken over $t \sim \mathcal{U}([0,T])$ and $(\mathbf{X}_0, \mathbf{X}_t)$. For an arbitrarily flexible score network, the global minimiser $\theta^\star = \arg\min_\theta \mathcal{L}(\theta)$ satisfies $s_{\theta^\star}(t,\cdot) = \nabla \log p_t$.

## 3 Inequality-Constrained Diffusion Models

We are now ready to introduce our methodology to deal with manifolds defined via *inequality constraints* (1). In Section 3.1, we propose a Riemannian diffusion model endowed with a metric induced by a log-barrier potential. In Section 3.2, we introduce a *reflected* diffusion model. While both models extend classical diffusion models to inequality-constrained settings, they exhibit very different behaviours. We discuss their practical differences in Section 5.1.

### 3.1 Log-barrier diffusion models

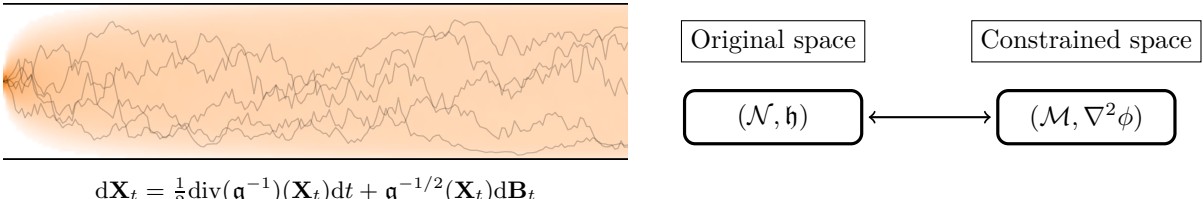

$$\mathrm{d}\mathbf{X}_t = \tfrac{1}{2}\text{div}(\mathfrak{g}^{-1})(\mathbf{X}_t)\mathrm{d}t + \mathfrak{g}^{-1/2}(\mathbf{X}_t)\mathrm{d}\mathbf{B}_t$$

Figure 3: Convergence of the Barrier Langevin dynamics on the unit interval to the uniform distribution.

Figure 4: Illustrative diagram of the barrier method and the change of metric.

**Barrier Langevin dynamics.** Barrier methods work by constructing a smooth potential $\phi: \mathcal{M} \to \mathbb{R}$ such that it blows up on the boundary of a desired set, see Nesterov et al. (2018). Such potentials form the basis of interior point methods in convex optimisation (Boyd et al., 2004). Of these functions, the *logarithmic barrier* is the most popular among practitioners (Lee & Vempala, 2017). For a convex polytope $\mathcal{M}$ defined by the constraints $\mathrm{A}x < b$, with $\mathrm{A} \in \mathbb{R}^{m \times d}$ and $b \in \mathbb{R}^m$, the logarithmic barrier $\phi: \mathcal{M} \to \mathbb{R}_+$ is given for any $x \in \mathcal{M}$ by

$$\phi(x) = -\sum_{i=1}^m \log(\langle \mathrm{A}_i, x \rangle - b_i). \tag{5}$$

Assuming that $\|\mathrm{A}_i\| = 1$, we have that for any $x \in \mathcal{M}$, $\phi(x) = -\sum_{i=1}^m \log(d(x, \partial\mathcal{M}_i))$, where $\partial\mathcal{M}_i = \{x \in \mathbb{R}^d : \langle \mathrm{A}_i, x \rangle = b\}$. More generally we can define for any $x \in \mathcal{M}$

$$\phi(x) = -\sum_{i=1}^m \log(d(x, \partial\mathcal{M}_i))$$

where $d(x, \partial\mathcal{M}_i)$ computes the minimum geodesic distance from $x$ to the boundary $\partial\mathcal{M}$. In general this is a highly non-trivial optimization problem, contrary to the polytope case which admits a simple closed form.

While developed and most commonly used in optimisation, barrier methods can also be used for sampling (Lee & Vempala, 2017). The core idea of barrier methods is to 'warp the geometry' of the constrained space, stretching it as the process approaches the boundary so that it never hits it, hence bypassing the need to explicitly deal with the boundary. Assuming $\phi$ to be strictly convex and smooth, its Hessian $\nabla^2\phi$ is positive definite and thus defines a valid Riemannian metric on $\mathcal{M}$. The formal approach to 'warping the geometry' of the convex space with the boundary is to endow $\mathcal{M}$ with the Hessian as a Riemannian metric $\mathfrak{g} = \nabla^2\phi$, making it into a Hessian Manifold, see Shima & Yagi (1997). In the special case where the barrier is given by (5), we get for any $x \in \mathcal{M}$

$$\mathfrak{g}(x) = \mathrm{A}^\top S^{-2}(x)\mathrm{A} \text{ with } S(x) = \text{diag}(b_i - \langle \mathrm{A}_i, x \rangle)_i.$$

Equipped with this Riemannian metric, we consider the following Langevin dynamics as a forward process

$$d\mathbf{X}_t = \tfrac{1}{2}\mathrm{div}(\mathfrak{g}^{-1})(\mathbf{X}_t)dt + \mathfrak{g}(\mathbf{X}_t)^{-\frac{1}{2}}d\mathbf{B}_t, \tag{6}$$

with $\mathrm{div}(F)(x) \triangleq (\mathrm{div}(F_1)(x), \ldots, \mathrm{div}(F_d)(x))^\top$, for any smooth $F : \mathbb{R}^d \to \mathbb{R}^d$. Under mild assumptions on $\mathcal{M}$, $\mathrm{div}(\mathfrak{g}^{-1})$ and $\mathfrak{g}^{-1}$ we get that $(\mathbf{X}_t)_{t\geq 0}$ is well-defined and for any $t \geq 0$, $\mathbf{X}_t \in \mathcal{M}$. In particular, for any $t \geq 0$, $\mathbf{X}_t$ does not reach the boundary. This stochastic process was first proposed by Lee & Vempala (2017) in the context of efficient sampling from the uniform distribution over a polytope. Under similar conditions, $\mathbf{X}_t$ admits a density $p_t$ w.r.t. the Lebesgue measure and we have that $\partial_t p_t = \tfrac{1}{2}\mathrm{Tr}(\mathfrak{g}^{-1}\nabla^2 p_t)$. In addition, $(\mathbf{X}_t)_{t\geq 0}$ is irreducible. Hence, assuming that $\mathcal{M}$ is compact, the uniform distribution on $\mathcal{M}$ is the unique invariant measure of the process $(\mathbf{X}_t)_{t\geq 0}$ and $(\mathbf{X}_t)_{t\geq 0}$ converges to the uniform distribution in some sense. We refer the reader to Appendix C for a proof of these results.

**Time-reversal.** Assuming that $\mathfrak{g}^{-1}$ and its derivative are bounded on $\mathcal{M}$, the time-reversal of (6) is given by Cattiaux et al. (2021), in particular we have

$$\begin{aligned} d\overleftarrow{\mathbf{X}}_t &= [-\tfrac{1}{2}\mathrm{div}(\mathfrak{g}^{-1}) + \mathrm{div}(\mathfrak{g}^{-1}) + \mathfrak{g}^{-1}\nabla\log p_{T-t}](\overleftarrow{\mathbf{X}}_t)dt + \mathfrak{g}(\overleftarrow{\mathbf{X}}_t)^{-\frac{1}{2}}d\mathbf{B}_t, \\ &= \left[\tfrac{1}{2}\mathrm{div}(\mathfrak{g}^{-1}) + \mathfrak{g}^{-1}\nabla\log p_{T-t}\right](\overleftarrow{\mathbf{X}}_t)dt + \mathfrak{g}(\overleftarrow{\mathbf{X}}_t)^{-\frac{1}{2}}d\mathbf{B}_t. \end{aligned} \tag{7}$$

$\overleftarrow{\mathbf{X}}_0$ is initialised with the uniform distribution on $\mathcal{M}$ (which is close to the one of $\mathbf{X}_T$ for large $T$).

The estimation of the score term $\nabla\log p_t$ is done by minimising the ism loss function (4). We refer to Section 3.3 for details on the training and parameterisation.

**Sampling.** Sampling from the forward (6) and backward (7) processes, once the score is learnt, requires a discretisation scheme. We use Geodesic Random Walks (GRW) (Jørgensen, 1975) for this purpose, see Algorithm 1. This discretisation is a generalisation of the Euler-Maruyama discretisation of SDE in Euclidean spaces, where the + operator is replaced by the exponential mapping on the manifold, computing the geodesics.

---

**Algorithm 1** *Geodesic Random Walk.* Discretisation of the SDE $d\mathbf{X}_t = d(t, \mathbf{X}_t)dt + d\mathbf{B}_t$.

---

**Require:** $T$ (simulation time), $N$ (number of steps), $X_0^\gamma$ (initial point), $d$ (drift function)
  $\gamma = T/N$
  **for** $k \in \{0, \ldots, N-1\}$ **do**
    $Z_{k+1} \sim \mathrm{N}(0, \mathrm{Id})$
    $W_{k+1} = \gamma d(k\gamma, X_k) + \sqrt{\gamma}Z_{k+1}$
    $X_{k+1}^\gamma = \exp_{X_k}[W_{k+1}] \approx \mathrm{proj}_{\mathcal{M}}(X_k + W_{k+1})$
  **return** $\{X_k\}_{k=0}^N$

---

However, contrary to De Bortoli et al. (2022), we will not have access explicitly to the exponential mapping of the Hessian manifold. Instead, we rely on an approximation, using a *retraction* (see Absil & Malick (2012); Boumal (2023) for a definition and alternative schemes).

## 3.2 Reflected diffusion models

Another approach to deal with the geometry of $\mathcal{M}$ is to use the standard metric $\mathfrak{h}$ and forward dynamics of $\mathcal{N}$ and constraining it to $\mathcal{M}$ by *reflecting* the process whenever it would encounter a boundary. We will first assume that $\mathcal{M}$ is compact and convex. To simplify the presentation, we focus on the Euclidean case $\mathcal{N} = \mathbb{R}^d$ with smooth boundary $\partial M$[4].

The key difference between this approach and the barrier approach is that in the reflected case we leave the geometry unchanged, so all we need to do is show that the dynamics induced by reflecting the forward process whenever it hits the boundary leads to an invariant distribution and admits a time-reversal.

---

[4]We refer to Appendix G for a definition of smooth boundary.

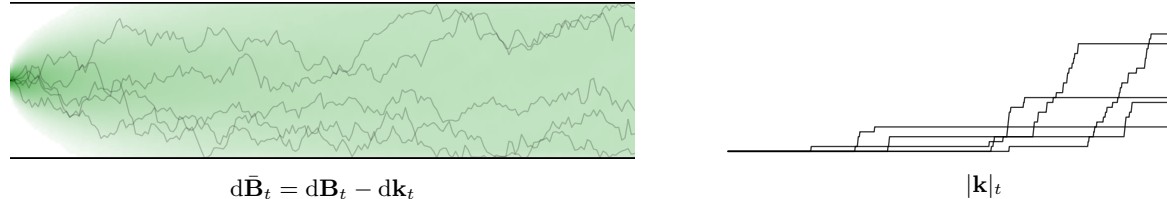

$$\mathrm{d}\bar{\mathbf{B}}_t = \mathrm{d}\mathbf{B}_t - \mathrm{d}\mathbf{k}_t \qquad\qquad\qquad |\mathbf{k}|_t$$

Figure 5: *Left:* Convergence of the reflected Brownian motion on the unit interval to the uniform distribution. *Right:* Value of $|\mathbf{k}|_t$ for the trajectory samples on the left through time.

It is worth noting that while barrier approaches have received considerable theoretical attention in the sampling literature (Lee & Vempala, 2017; Noble et al., 2022), reflected methods have remained comparatively undeveloped from the methodological and practical point of view. In the next section, we recall the basics of reflected stochastic processes.

**Skorokhod problem.** The reflected Brownian motion is defined as the solution to the Skorokhod problem. Roughly speaking a solution to the Skorokhod problem consists of two coupled processes, $(\bar{\mathbf{B}}_t, \mathbf{k}_t)_{t\geq 0}$, such that $(\bar{\mathbf{B}}_t)_{t\geq 0}$ acts *locally* as a Euclidean Brownian motion $(\mathbf{B}_t)_{t\geq 0}$ and $\mathbf{k}_t$ compensates for the excursion of $(\bar{\mathbf{B}}_t)_{t\geq 0}$ so that $(\bar{\mathbf{B}}_t)_{t\geq 0}$ remains in $\mathcal{M}$. We say that $(\bar{\mathbf{B}}_t, \mathbf{k}_t)_{t\geq 0}$ is a solution to the *Skorokhod problem* (Skorokhod, 1961) if $(\mathbf{k}_t)_{t\geq 0}$ and $(\bar{\mathbf{B}}_t)_{t\geq 0}$ are two processes satisfying mild conditions, see Appendix D for a rigorous introduction, such that for any $t \geq 0$,

$$\bar{\mathbf{B}}_t = \bar{\mathbf{B}}_0 + \mathbf{B}_t - \mathbf{k}_t \in \mathcal{M}, \tag{8}$$

and $|\mathbf{k}|_t = \int_0^t \mathbf{1}_{\bar{\mathbf{B}}_s \in \partial\mathcal{M}} \mathrm{d}|\mathbf{k}|_s$, $\mathbf{k}_t = \int_0^t \mathbf{n}(\bar{\mathbf{B}}_s)\mathrm{d}|\mathbf{k}|_s$, where $(|\mathbf{k}|_t)_{t\geq 0}$ is the total variation of $(\mathbf{k}_t)_{t\geq 0}$ and we recall that $\mathbf{n}$ is the outward normal to $\mathcal{M}$[5]. When $(\bar{\mathbf{B}}_t)_{t\geq 0}$ hits the boundary, the condition $\mathbf{k}_t = \int_0^t \mathbf{n}(\bar{\mathbf{B}}_s)\mathrm{d}|\mathbf{k}|_s$, tells us that $-\mathbf{k}_t$ "compensates" for $\bar{\mathbf{B}}_t$ by pushing the process back into $\mathcal{M}$ along the inward normal $-\mathbf{n}$, while the condition $|\mathbf{k}|_t = \int_0^t \mathbf{1}_{\bar{\mathbf{B}}_s \in \partial\mathcal{M}} \mathrm{d}|\mathbf{k}|_s$ can be interpreted as $\mathbf{k}_t$ being constant when $(\bar{\mathbf{B}}_t)_{t\geq 0}$ does not hit the boundary. As a result $(\bar{\mathbf{B}}_t)_{t\geq 0}$ can be understood as the continuous-time counterpart to the reflected Gaussian random walk. The process $(\mathbf{k}_t)_{t\geq 0}$ can be related to the notion of *local time* (Revuz & Yor, 2013) and quantifies the amount of time $(\bar{\mathbf{B}}_t)_{t\geq 0}$ spends at the boundary $\partial\mathcal{M}$. Lions & Sznitman (1984, Theorem 2.1) ensure the existence and uniqueness of a solution to the Skorokhod problem. One key observation is that the event $\{\bar{\mathbf{B}}_t \in \partial\mathcal{M}\}$ has probability zero (Harrison & Williams, 1987, Section 7, Lemma 7). As in the *unconstrained* setting, one can describe the dynamics of the density of $\bar{\mathbf{B}}_t$.

**Proposition 3.1** (Burdzy et al. (2004))**.** *For any $t > 0$, the distribution of $\bar{\mathbf{B}}_t$ admits a density w.r.t. the Lebesgue measure denoted $p_t$. In addition, we have for any $x \in \mathrm{int}(\mathcal{M})$ and $x_0 \in \partial\mathcal{M}$*

$$\partial_t p_t(x) = \tfrac{1}{2}\Delta p_t(x), \quad \partial_{\mathbf{n}} p_t(x_0) = 0, \tag{9}$$

*where we recall that $\mathbf{n}$ is the outward normal to $\mathcal{M}$.*

Note that contrary to the unconstrained setting, the heat equation has *Neumann* boundary conditions. Similarly to the compact Riemannian setting (Saloff-Coste, 1994) it can be shown that the reflected Brownian motion converges to the uniform distribution on $\mathcal{M}$ exponentially fast (Loper, 2020; Burdzy et al., 2006), see Section 3.2. Hence, $(\bar{\mathbf{B}}_t)_{t\geq 0}$ is a candidate for a forward noising process in the context of diffusion models.

**Time-reversal.** In order to extend the diffusion model approach to the reflected setting, we need to derive a *time-reversal* for $(\bar{\mathbf{B}}_t)_{t\in[0,T]}$. Namely, we need to characterise the evolution of $(\overleftarrow{\mathbf{X}}_t)_{t\in[0,T]} = (\bar{\mathbf{B}}_{T-t})_{t\in[0,T]}$. It can be shown that the time-reversal of $(\bar{\mathbf{B}}_t)_{t\in[0,T]}$ is also the solution to a Skorokhod problem.

---

[5]We extend the normal $\mathbf{n}$ to the whole space by letting $\mathbf{n}(x) = 0$ if $x \notin \partial\mathcal{M}$.

---

**Algorithm 2** *Reflected step algorithm.* The algorithm operates by repeatedly taking geodesic steps until one of the constraints is violated, or the step is fully taken. Upon hitting the boundary we parallel transport the tangent vector to the boundary and then reflect it against the boundary. We then start a new geodesic from this point in the new direction. The $\arg\mathrm{intersect}_t$ function computes the distance one must travel along a geodesic in direction $\boldsymbol{s}$ til constraint $f_i$ is intersected. For a discussion of paralleltransport, $\exp_{\mathfrak{g}}$ and reflect please see Appendix A.

---

**Input:** $x \in \mathcal{M}$, $\boldsymbol{v} \in \mathrm{T}_x\mathcal{M}$, $\{f_i\}_{i\in\mathcal{I}}$
  $\ell \leftarrow \|\boldsymbol{v}\|_{\mathfrak{g}}$
  $\boldsymbol{s} \leftarrow \boldsymbol{v}/\|\boldsymbol{v}\|_{\mathfrak{g}}$
  **while** $\ell \geq 0$ **do**
    $d_i = \arg\mathrm{intersect}_t\big[\exp_{\mathfrak{g}}(x, t\boldsymbol{s}), f_i\big]$
    $i \leftarrow \arg\min_i\ d_i\ s.t.\ d_i > 0$
    $\alpha \leftarrow \min(d_i, \ell)$
    $x' \leftarrow \exp_{\mathfrak{g}}(x, \alpha\boldsymbol{s})$
    $\boldsymbol{s} \leftarrow \mathrm{paralleltransport}_{\mathfrak{g}}(x, \boldsymbol{s}, x')$
    $\boldsymbol{s} \leftarrow \mathrm{reflect}(\boldsymbol{s}, f_i)$
    $\ell \leftarrow \ell - \alpha$
    $x \leftarrow x'$
  **return** $x$

---

**Algorithm 3** *Reflected Random Walk.* Discretisation of the SDE $\mathrm{d}\mathbf{X}_t = b(t, \mathbf{X}_t)\mathrm{d}t + \mathrm{d}\mathbf{B}_t - \mathrm{d}\mathbf{k}_t$.

---

**Require:** $T$ (simulation time), $N$ (number of steps), $X_0^{\gamma}$ (initial point), $\{f_i\}_{i\in\mathcal{I}}$ (boundary functions)
  $\gamma = T/N$
  **for** $k \in \{0, \dots, N-1\}$ **do**
    $Z_{k+1} \sim \mathrm{N}(0, \mathrm{Id})$
    $X_{k+1}^{\gamma} = \mathrm{ReflectedStep}[X_k^{\gamma}, \sqrt{\gamma}Z_{k+1}, \{f_i\}_{i\in\mathcal{I}}]$
  **return** $\{X_k^{\gamma}\}_{k=0}^N$

---

**Theorem 3.2.** *There exist $(\overleftarrow{\mathbf{k}}_t)_{t\geq 0}$ a bounded variation process and a Brownian motion $(\mathbf{B}_t)_{t\geq 0}$ such that*

$$\overleftarrow{\mathbf{X}}_t = \overleftarrow{\mathbf{X}}_0 + \mathbf{B}_t + \int_0^t \nabla\log p_{T-s}(\overleftarrow{\mathbf{X}}_s)\mathrm{d}s - \overleftarrow{\mathbf{k}}_t.$$

*In addition, for any $t \in [0, T]$ we have*

$$|\overleftarrow{\mathbf{k}}|_t = \int_0^t \mathbf{1}_{\overleftarrow{\mathbf{X}}_s \in \partial\mathcal{M}}\mathrm{d}|\overleftarrow{\mathbf{k}}|_s, \quad \overleftarrow{\mathbf{k}}_t = \int_0^t \mathbf{n}(\overleftarrow{\mathbf{X}}_s)\mathrm{d}|\overleftarrow{\mathbf{k}}|_s.$$

The proof, see Appendix G, follows Petit (1997) which provides a time-reversal in the case where $\mathcal{M}$ is the positive orthant. It is based on an extension of Haussmann & Pardoux (1986) to the reflected setting, with a careful handling of the boundary conditions. In particular, contrary to Petit (1997), we do not rely on an explicit expression of $p_t$ but instead use the intrinsic properties of $(\mathbf{k}_t)_{t\geq 0}$. Informally, Theorem 3.2 means that the process $(\overleftarrow{\mathbf{X}}_t)_{t\in[0,T]}$ satisfies

$$\mathrm{d}\overleftarrow{\mathbf{X}}_t = \nabla\log p_{T-t}(\overleftarrow{\mathbf{X}}_t)\mathrm{d}t + \mathrm{d}\mathbf{B}_t - \mathrm{d}\overleftarrow{\mathbf{k}}_t, \tag{10}$$

which echoes the usual time-reversal formula (2). In practice, in order to sample from $(\overleftarrow{\mathbf{X}}_t)_{t\in[0,T]}$, one needs to consider the *reflected* version of the *unconstrained* dynamics $\mathrm{d}\overleftarrow{\mathbf{X}}_t = \nabla\log p_{T-t}(\overleftarrow{\mathbf{X}}_t)\mathrm{d}t + \mathrm{d}\mathbf{B}_t$.

**Sampling.**  In practice, we approximately sample the reflected dynamics by considering the Markov chain given by Algorithm 3. We refer to Pacchiarotti et al. (1998) and Bossy et al. (2004) for weak convergence results on this numerical scheme in the Euclidean setting.

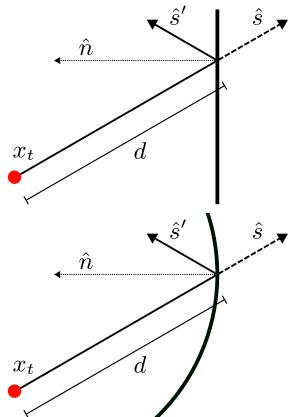

Figure 6: Reflection against a linear boundary. For a step $s$ with magnitude $|s|$ and direction $\hat{s}$ the distance to the boundary described by the normal $\hat{n}$ and offset $b$ is $d = \frac{\langle \hat{s}, x_t \rangle - b}{\langle \hat{s}, \hat{n} \rangle}$. The reflected direction is given by $\hat{s}' = \hat{s} - 2\langle \hat{s}, \hat{n} \rangle \hat{n}$.

Figure 7: Reflection against a spherical boundary. For a sphere of radius $r$, the distance to the boundary from $x_t$ in direction $s$ is given by $d = \frac{1}{2}(\langle \hat{s}, x_t \rangle^2 + 4(r^2 - \|x_t\|^2))^{1/2} - \frac{1}{2}\langle \hat{s}, x_t \rangle$. The normal at the intersection can be computed as the unit vector in the direction $-2(d\hat{s} + x_t)$, and then $\hat{s}'$ as above.

**Likelihood evaluation.**  In the case of a reflected Brownian motion, it is possible to compute an equivalent ODE in order to perform likelihood evaluation. The associated ODE was first derived in Lou & Ermon (2023). The form of the ODE and the proof that it remains in $\mathcal{M}$ are postponed to Appendix F.

### 3.3  Loss and score network parameterisation

In order to train log-barrier and reflected diffusion models we prove that we can use a *tractable* score matching loss in constrained manifolds. We will prove that the implicit score-matching loss leads to the recovery of the correct score when we have a boundary, so long as we enforce that the score is zero on the boundary (see Appendix E.1). This proof holds for both the log-barrier and the reflected process.

**Proposition 3.3.** *Let $s \in C^\infty([0, T] \times \mathbb{R}^d, \mathbb{R}^d)$ such that for any $x \in \partial\mathcal{M}$ and $t \geq 0$, $s_t(x) = 0$. Then, there exists $C > 0$ such that*

$$\mathbb{E}[\|\nabla \log p_t - s_t\|^2] = \mathbb{E}[\|s_t\|^2 + 2 \operatorname{div}(s_t)] + C,$$

*where $\mathbb{E}$ is taken over $\mathbf{X}_t \sim p_t$ and $t \sim \mathcal{U}([0, T])$.*

This result immediately implies we can optimise the score network using the ism loss function so long as we enforce a Neumann boundary condition. The estimation of the score term $\nabla \log p_t$ is also done by minimising the ism loss function (4).

The score term $\nabla \log p_{T-t}$ appearing in both time-reversal processes (7) and (10) is intractable. It is thus approximated with a *score network* $s_\theta(t, \cdot) \approx \nabla \log p_t$. We use a multi-layer perceptron architecture with sin activation functions, see Appendix J for more details on the experimental setup. Due to boundary condition (9), the normal component of the score is zero at the boundary in the reflected case. A similar result holds in the log-barrier setting. This is additionally required for our proof of the ism loss.

Following Liu et al. (2022), we can accommodate this in our score parameterisation by additionally scaling the score output of the neural network by a monotone function $h(d(x, \partial\mathcal{M}))$ where $d$ is the distance from $x$ to the boundary and $h(0) = 0$. In particular, we use a clipped ReLU function: $s_\theta(t, x) = \min(1, \operatorname{ReLU}(d(x, \partial\mathcal{M}) - \delta)) \cdot \operatorname{NN}_\theta(t, x)$ with $\delta > 0$ a threshold so the model is forced to be zero "close" to the boundary as well as exactly on the boundary, see Appendix J for an illustration. The inclusion of this scaling function is necessary to produce reasonable results as we show in Appendix E.2. The weighting function in (4) is set to $\lambda_t = (t+1)$.

The forward processes (6) and (8) for the barrier and reflected methods cannot be sampled in closed form, so at training time samples from the conditional marginals $p(\mathbf{X}_t | \mathbf{X}_0)$ are obtained by discretising these processes. As to take the most of this computational overhead, we use several samples from the discretised forward trajectory $(\mathbf{X}_{t_1}, \cdots, \mathbf{X}_{t_k} | \mathbf{X}_0)$ instead of only using the last sample.

## 4 Related Work

**Sampling on constrained manifolds.** Sampling from a distribution on a space defined by a set of constraints is an important ingredient in several computational tasks, such as computing the volume of a polytope (Lee & Vempala, 2017). Incorporating such constraints within MCMC algorithms while preserving fast convergence properties is an active field of research (Kook et al., 2022; Lee & Vempala, 2017; Noble et al., 2022). In this work, we are interested in sampling from the uniform distribution defined on the constrained set in order to define a proper *forward process* for our diffusion model. Log-barrier methods such as the Dikin walk or Riemannian Hamiltonian Monte Carlo (Kannan & Narayanan, 2009; Lee & Vempala, 2017; Noble et al., 2022) change the geometry of the underlying space and define stochastic processes which never violate the constraints, see Kannan & Narayanan (2009) and Noble et al. (2022) for instance. If we keep the Euclidean metric, then *unconstrained* stochastic processes might not be well-defined for all times. To counter this effect, it has been proposed to *reflect* the Brownian motion (Williams, 1987; Petit, 1997; Shkolnikov & Karatzas, 2013). Finally, we also highlight hit-and-run approaches (Smith, 1984; Lovász & Vempala, 2006), which generalise Gibbs' algorithm and enjoy fast convergence properties provided that one knows how to sample from the one-dimensional marginals.

**Diffusion models on manifolds.** De Bortoli et al. (2022) extended the work of Song et al. (2021) to Riemannian manifolds by defining forward and backward stochastic processes in this setting. Concurrently, a similar framework was introduced by Huang et al. (2022), extending the maximum likelihood approach of Huang et al. (2021). Existing applications of denoising diffusion models on Riemannian manifolds have been focused on well-known manifolds for which one can find metrics so that the framework of De Bortoli et al. (2022) applies. In particular, on compact Lie groups, geodesics and Brownian motions can be defined in a canonical manner. Their specific structure can be leveraged to define efficient diffusion models (Yim et al., 2023). Leach et al. (2022) define diffusion models on SO(3) for rotational alignment, while Jing et al. (2022) consider the product of tori for molecular conformer generation. Corso et al. (2022) use diffusion models on $\mathbb{R}^3 \times \mathrm{SO}(3) \times \mathrm{SO}(2)$ for protein docking applications. RFDiffusion (Watson et al., 2022) and FrameDiff also incorporate SE(3) diffusion models. Finally, Urain et al. (2022) introduce a methodology for SE(3) diffusion models with applications to robotics.

**Comparison with Lou & Ermon (2023).** We now discuss a few key differences between our work and the reflected diffusion models presented in Lou & Ermon (2023). First, in the hypercube setting, our methodologies are identical from a theoretical viewpoint. The main difference is that Lou & Ermon (2023) use an approxiamted version of the DSM loss (3), whereas we rely on the ISM loss (4). Lou & Ermon (2023) exploit the specific factorised structure of the hypercube to make the DSM loss tractable, leading to significant practical advantages. These can however be directly employed in our framework for reflected models, and also in the log-barrier setting. Second, in our work, we target scientific applications where the underlying geometry is not Euclidean, whereas Lou & Ermon (2023) focus on the case where the constrained domain of interest is the hypercube (a subset of Euclidean space) with image applications, or where the domain can be easily projected into the hypercube, such as the simplex. Our approach is designed to handle a wider range of settings, such as non-convex polytopes in Euclidean space, or non-Euclidean geometries.

## 5 Experimental results

To demonstrate the practical utility of the constrained diffusion models introduced in Section 3, we evaluate them on a series of increasingly difficult synthetic tasks on convex polytopes, including the hypercube and the simplex, in Section 5.1. We then highlight their applicability to real-world settings by considering two problems from robotics and protein design. In particular, we show that our models are able to learn distributions over the space of $d \times d$ symmetric positive definite (SPD) matrices $S_{++}^d$ under trace constraints in Section 5.2—a setting that is essential to describing and controlling the motions and exerted forces of robotic platforms (Jaquier et al., 2021). In Section 5.3, we use the parametrisation introduced in Han & Rudolph (2006) to map the problem of modelling the conformational ensembles of proteins under positional constraints on their endpoints to the product manifold of a convex polytope and a torus. The code is available here. We refer the reader to Appendix J for more details on the experimental setup.

Table 1: MMD metrics between samples from synthetic distributions and trained constrained and unconstrained (Euclidean) diffusion models. Means and confidence intervals are computed over 5 different runs.

| Space | $d$ | Log-barrier | | Reflected | | Euclidean | |
|---|---|---|---|---|---|---|---|
| | | MMD | % in $\mathcal{M}$ | MMD | % in $\mathcal{M}$ | MMD | % in $\mathcal{M}$ |
| $[-1,1]^d$ | 2 | $.066_{\pm.006}$ | 100.0 | $.055_{\pm.015}$ | 100.0 | $.062_{\pm.011}$ | 98.8 |
| | 3 | $.209_{\pm.077}$ | 100.0 | $.080_{\pm.004}$ | 100.0 | $.076_{\pm.004}$ | 98.5 |
| | 10 | $.330_{\pm.004}$ | 100.0 | $.313_{\pm.048}$ | 100.0 | $.081_{\pm.005}$ | 96.4 |
| $\Delta^d$ | 2 | $.050_{\pm.012}$ | 100.0 | $.043_{\pm.002}$ | 100.0 | $.055_{\pm.013}$ | 96.4 |
| | 3 | $.238_{\pm.009}$ | 100.0 | $.181_{\pm.007}$ | 100.0 | $.068_{\pm.014}$ | 96.3 |
| | 10 | $.275_{\pm.015}$ | 100.0 | $.290_{\pm.009}$ | 100.0 | $.060_{\pm.003}$ | 92.6 |

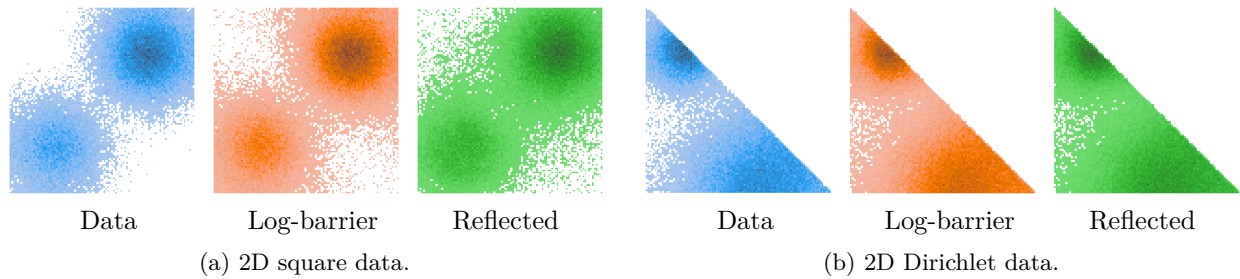

| Data | Log-barrier | Reflected | Data | Log-barrier | Reflected |
|---|---|---|---|---|---|

(a) 2D square data.            (b) 2D Dirichlet data.

Figure 8: Histograms of samples from the data distribution and from trained constrained diffusion models.

## 5.1 Method characterisation on convex polytopes

First, we aim to assess the empirical performance of our methods on constrained manifolds of increasing dimensionality. For this, we focus on polytopes and construct synthetic datasets that represent bimodal distributions. In particular, we investigate two specific instances of polytopes: hypercubes and simplices. In Appendix J.1 we also present results on the Birkhoff polytope. We quantify the performance of each model via the Maximum Mean Discrepancy (MMD) (Gretton et al., 2012), which is a kernel-based metric between distributions. We present a qualitative comparison of the logarithmic barrier and reflected Brownian motion models in Figure 8, and observe that both methods recover the data distribution on the two-dimensional hypercube and simplex, although the reflected method seems to produce a better fit. In Table 1, we report the MMD between the data distribution and the learnt diffusion models. Here, we similarly observe that the reflected method consistently yields better results than the log-barrier one.

We additionally compare both of these models to a set of unconstrained Euclidean diffusion models, noting that they are outperformed by the constrained models in lower dimensions, but generate better results in higher dimensions. There are a number of potential explanations for this: First, we note that the mixture of Normal distributions we use as a synthetic data-generating process places significantly less probability mass near the boundary as its dimensionality increases, more closely resembling an unconstrained mixture distribution that is easier for the Euclidean diffusion models to learn, while posing a challenge to the log-barrier and reflected diffusion models that initialise at the uniform distribution within the constraints. Additionally, we note that the design space and hyperparameters used for all experiments were informed by best practices for Euclidean models that may be suboptimal for the more complex dynamics of constrained diffusion models.

## 5.2 Modelling robotic arms under force and velocity constraints

Accurately determining and controlling the movement and exerted forces of robotic platforms is a fundamental problem in many real-world robotics applications. A kinetostatic descriptor that is commonly used to quantify the ability of a robotic arm to move and apply forces along certain dimensions is the so-called manipulability ellipsoid $E \in \mathbb{R}^d$ (Yoshikawa, 1985), which is naturally described as a symmetric positive definite (SPD)

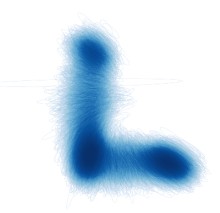

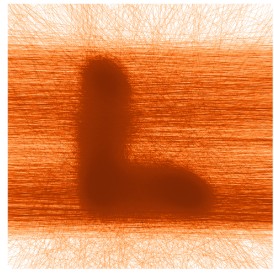

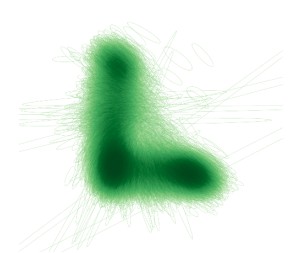

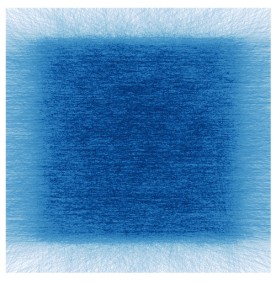

(a) Samples from the data distribution.

(b) Samples from our log-barrier diffusion model.

(c) Samples from our reflected diffusion model.

(d) Samples from the uniform distribution.

Figure 9: Samples in $S_{++}^2 \times \mathbb{R}^2$ from (a) the data distribution, (b) our log-barrier diffusion model, (c) our reflected diffusion model and (d) the uniform distribution. Each sample is visualised as the manipulability ellipsoid encoded by the SPD matrix $M \in S_{++}^2$ placed at the corresponding location in $\mathbb{R}^2$. Additional results and full correlation plots are postponed to Appendix J.2.

matrix $M \in \mathbb{R}^{d \times d}$ (Jaquier et al., 2021). The manifold of such $d \times d$ SPD matrices, denoted as $S_{++}^d$, is defined as the set of matrices $\{x^\top Mx \geq 0, \ x \in \mathbb{R}^d : M \in \mathbb{R}^{d \times d}\}$. In many practical settings, it may be desirable to constrain the volume of $E$ to retain flexibility or limit the magnitude of an exerted force, which can be expressed as an upper bound on the trace of M, i.e. as the inequality constraint $\sum_{i=1}^d M_{ii} < C$ with $C > 0$. Constraining the rest of the entries of the matrix to ensure it is SPD is non-trivial. Alternatively, we can parameterise the SPD matrices via their Cholesky decomposition. Each SPD matrix has a unique decomposition of the form $M = LL^\top$, where L is a lower triangular matrix with strictly positive diagonal (Golub & Van Loan, 2013, p.143). Constraining the entries of this matrix simply requires ensuring the diagonal is positive. The trace of the SPD matrix is given by $\text{Tr}(M) = \text{Tr}(LL^\top) = \sum_{ij} L_{ij}^2$, and results in the constraint on the entries of L to live in a ball of radius $C$. We additionally model the two-dimensional position of the arm. In summary, the space over which we parameterise the diffusion models is defined as $\{L \in \mathbb{R}^{d(d+1)/2} : L_{i,i} > 0, \sum_{i,j} L_{i,j}^2 < C\} \times \mathbb{R}^2$. Under the Euclidean metric, we can apply both our log-barrier and reflected approaches. The positive diagonal (linear) constraint is handled similarly to the polytope setting. The reflection on the ball boundary is defined and illustrated on Figures 6 and 7.

Using this framework, we model the datasets presented in Jaquier et al. (2021) (see Appendix H for full experimental details). The joint distribution over the SPD matrices (represented as ellipsoids) and their positions is presented in Figure 9. We qualitatively observe that the reflected method is able to model the joint data distribution better than the log-barrier one. This is reflected by an MMD of 0.161 and 0.247, respectively.

### 5.3 Modelling protein loops with anchored endpoints

Modelling the conformational ensembles of proteins is an important task in the field of molecular biology, particularly in the context of bioengineering and drug discovery. In many data-scarce practical settings such as antibody or enzyme design, it is often unnecessary or even undesirable to model the structure of an entire protein, as researchers are primarily interested in specific functional sites with distinct biochemical properties. However, generating conformational ensembles for such substructural elements necessitates positional constraints on their endpoints to ensure that they can be accommodated by the remaining scaffold.

This problem can be reformulated as modelling a spatial chain with spherical joints and fixed end points. Following the framework outlined in Han & Rudolph (2006), we parametrise the conformations of such chains with $d$ fixed-length links and arbitrary end-point constraints as the product of a convex polytope $\mathbb{P}$ and torus $\mathbb{T}$: $\mathbb{P}^{(d-3)} \times \mathbb{T}^{(d-2)}$. The essential idea of this parameterisation is to fix one end point as an "anchor" and model the chain as the series of triangles formed by the anchor and each pair of adjacent joints in the chain. A point in the polytope corresponds to the lengths of the diagonals of these triangles, and a point in the torus to the angles between each pair of subsequent triangles. See Figure 10 for an illustration and Appendix I.1 for a full description.

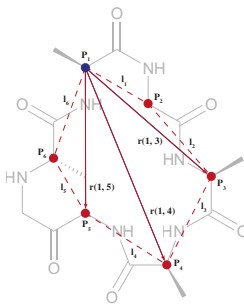
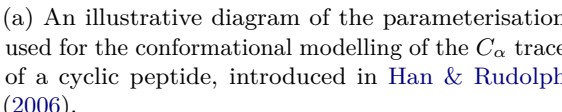

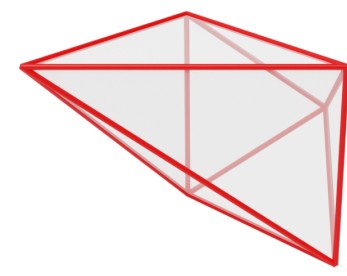

(a) An illustrative diagram of the parameterisation used for the conformational modelling of the $C_\alpha$ trace of a cyclic peptide, introduced in Han & Rudolph (2006).

(b) The convex polytope constraining the diagonals of the triangles for the given bond lengths in the illustrated molecule. The total design space is the product of this polytope with the 4D flat torus.

Figure 10: Illustrations of the parametrisation used to model distribution of polypeptide backbone conformations under anchor point distance constraints as the product of a convex polytope $\mathbb{P}$ and torus $\mathbb{T}$: $\mathbb{P}^3 \times \mathbb{T}^4$.

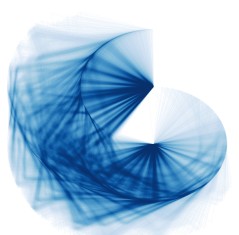
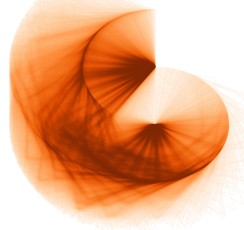
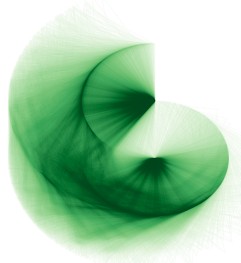
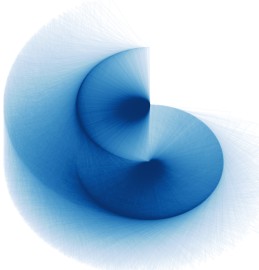

(a) Samples from the data distribution.

(b) Samples from our log-barrier diffusion model.

(c) Samples from our reflected diffusion model.

(d) Samples from the uniform distribution.

Figure 11: Planar projection of the modelled $C_\alpha$ chains from (a) the training dataset, (b) our log-barrier diffusion model, (c) our reflected diffusion model and (d) the uniform distribution. Additional results and full correlation plots are postponed to Appendix J.3.

Using this framework, we model the conformational landscape of the cyclic peptide c-AAGAGG, consisting of a circular polypeptide chain with coinciding endpoints. We generate $10^6$ backbone conformations using tools from molecular dynamics (Eastman et al., 2017; Hornak et al., 2006) and divide them into training and evaluation datasets, (see Appendix I.2 for full experimental details). Drawing on the definition above, the space on which we effectively parameterise our constrained diffusion models for a circular polypeptide chain of length $d = 6$ is given by the product manifold $\mathbb{P}^3 \times \mathbb{T}^4$.

To learn a distribution over this space, we leverage the methodology introduced in Section 3 for the polytope component $\mathbb{P}^3$ and in De Bortoli et al. (2022) for the torus component $\mathbb{T}^4$. A qualitative comparison of samples from the data distribution, our log-barrier and reflected diffusion models, and the uniform distribution is presented in Figures 11a to 11d. For enhanced visual clarity, we project the modelled spatial chain onto the 2D plane by removing the (unconstrained) torus component of the product manifold and only plotting the planar chains encoded by the (constrained) polytope component (a correlation plot of the full product manifold is presented in Figure 25).

It is apparent that the data distribution is highly multimodal, encompassing a large number of locally optimal conformational clusters. Nevertheless, our reflected and log-barrier diffusion models are able to robustly approximate this energetic landscape, producing samples that reflect key conformational states and producing comparable MMD metrics of $0.032_{\pm 0.021}$ and $0.032_{\pm 0.001}$, respectively. As a point of comparison, the uniform distribution on the polytope-torus product has an MMD of $0.112_{\pm 0.001}$.

## 6 Discussion

Learning complex distributions whose support is confined to constrained spaces is a crucial task with many applications in the natural and engineering sciences, including computational statistics (Morris, 2002), robotics (Han & Rudolph, 2006), quantum physics (Lukens et al., 2020) and computational biology (Thiele et al., 2013). In this work we extend continuous diffusion models to this setting, proposing two complementary approaches—one based on log-barrier methods and the other on the reflected Brownian motion. For both methods, we derive the time-reversal formula, propose discretisation schemes and extend the score-matching toolbox. We demonstrate the utility of our methods on a range of synthetic and real-world tasks, including the constrained conformational modelling of proteins and robotic arms, and find that reflected methods, while enjoying fewer theoretical guarantees than their log-barrier counterparts, often yield preferable results.

We conclude by highlighting important directions of future research. First, the computational cost of performing the reflection when discretising the reflected Brownian motion is high. Finding numerically efficient approximations of the reflected process is therefore necessary to extend this methodology to very high dimensional settings. Second, the retraction used in place of the exponential map for the barrier method leads to a high number of discretisation steps to ensure a good approximation. Designing a faster forward process for the log-barrier method is key to targeting more complex distributions.

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
