# OpenReview forum: "Diffusion Models for Constrained Domains"
_TMLR — Accepted by TMLR_

### Review · Reviewer_x71F · 2023-05-22

**Summary Of Contributions:**

The paper introduces diffusion models on constrained domains. While diffusion models have already been explored in the manifold setting, the case of bounded domains remains open, as one can show that the forward process a.s. escapes the domain. Therefore, two approaches are introduced: the log barrier and the reflected diffusion model. The log barrier defines a potential which penalizes if the process gets close to the boundary. The reflected diffusion model establishes another (local) BM which stays inside the domain and admits a decomposition into a standard BM and a bounded variation process. This allows to define a reverse process.

The two approaches are underlined by rigorous theorems that prove that the forward processes admit a stationary distribution, the corresponding statements for the reverse SDEs and some experiments. The first experiment is on toy data and the last two are from robotic modelling and molecular chemistry.

**Audience:**

Yes

**Broader Impact Concerns:**

As this is mainly a theoretical paper, I do not see any negative implications and no need for a broader impact statement.

**Claims And Evidence:**

Yes

**Requested Changes:**

Please clarify the weaknesses. The most pressing issue in my opinion is the lack of comparison with a "standard" diffusion model (trained with a DSM loss).

**Strengths And Weaknesses:**

The article has the following strengths.

- The article is generally well written, although the topic is very technical. However one needs a solid understanding in stochastic analysis and differential geometry to grasp it. I think this is due to the nature of the topic and inevitable.

-  Related work is covered well.

 - The plenty visualizations are very helpful for understanding the concepts and are well executed.

 - The article is building a rigorous theory of both approaches. All the necessities of defining a diffusion model (establishing a forward process, stationary distribution, reverse SDE) are proved in a lot of detail. One has to refer to the appendix at several points for a solid understanding.

- The two experiments are interesting applications where the data is in a constrained domain and there is a need for adapted algorithms to fit these constraints.

I will list the major weaknesses here:

 - In the concurrent work [Lou, Ermon] the ism loss approach is not taken. They argue that the ism loss will scale poorly to the high dimensional setting since almost all mass is concentrated close to the boundary (in the case of $[0,1]^d$) and this loss is downweighted due to their slightly differing derivation. How do you think this loss behaves in high dimensions? Have you tried the classical image generation task?

 - It is hard to interpret the experimental results, as they require some domain knowledge. While this is fine, I would like to see a standard diffusion model (in Euclidean space) trained on some of those to get a feeling of how the two approaches perform. In particular if a metric like MMD is chosen, one can even compare the result since standard diffusion should approximate the data manifold as well.

- The protein loop experiment is missing quantitative evaluations. I think it is not too high dimensional in order to take MMD (or even NLL) to compare the different methods. This is important because visually the log barrier method looks closer to the ground truth to me, however this would contradict the conclusions of the preceding experiments.

Minor Weaknesses/Questions:

- The DSM loss is missing a minus. Furthermore for both the DSM and the ISM loss standard references should be cited in section 3 as well not only in the intro.

 - In section 3.3 the clipped network parameterization does not make sense. Do you mean minimum here?

 - In section 5.1 you write that one advantage of the log barrier method is the equivalent ODE. However in appendix F you derive this equivalent ODE in the reflected case. This seems to be a relict of an earlier version?

 - Proposition F.1 does not contain a mathematical statement, it is only assumptions (as far as I see). Also the proof of it needs some work. I dont understand how smoothness of the score implies that $Y_t$ is on the boundary of M.

  - In Lemma G.2 you use the surface measure on $\partial M$ without having introduced it.

 - For the MMD evaluations you should write out which kernels you use. Also how many samples are used to calculuate the MMD? How much do the results depend on the used kernel?

- The fact that the reflected Brownian motion converges to the uniform distribution on the manifold from [Loper,2020] only seems to hold on convex sets which is more specific than your definition of M. Please clarify if this only holds on convex sets.

 - The typesetting in E.1 does not look too nice.

 - [Ablin, 2022] is not the standard reference for retractions in differential geometry.

- Can you comment on the regularity of the score in the reflected case? Here, the density at time t is not given anymore by the convolution of the data density and a gaussian transition, but something more complicated (reflected Gaussian kernel)? (This is just a curiosity of me, not a weakness)

- The description of Algorithm 2 is too long. I would suggest putting a more detailed description of it into the appendix. Also the abstract notions of parallel transport,... could be described in a more computational way.

---

> ### Author Response · Authors · 2023-06-07
> **Response to Reviewer x71F (1 / 2)**
>
> We thank the reviewer for their insightful assessment of our paper. We address their concerns below.
>
> ## Major Concerns
> 1. We are currently implementing experiments on images, leveraging the factorization available for the hypercube for both the log-barrier and the reflected processes. We are unsure how much progress can be made on this in two weeks, but plan to test this on, in order, low dimension hypercubes (sanity check), MNIST, CIFAR 10, and see how far we can get before the end of the review period.
> 2. We have added the Euclidean evaluations for the simulated experiments. In lower dimensions we see that the constrained models do better; in higher dimensions the Euclidean model performs better. There are a number of confounding factors here. First of all the data generating process for the synthetic experiments is actually a mixture of normals constrained within the boundary, as displayed in the plots. In higher dimensions our simulated distributions place much less mass near the boundary and more closely resemble a mixture of normals. This is a very natural distribution for the Euclidean diffusion models to learn, and relatively more challenging for the log-barrier/reflected models which initialize at the uniform within the constraints. An alternative factor is that the dynamics of the Euclidean model remain simple in high dimensions, while the constrained processes become more complex in higher dimensions. Finally as the reviewer suggests it may be that the ism loss is causing problems for the constrained methods in higher dimensions. One more factor may be relevant: we did not significantly explore the design space for constrained diffusion models. We use all the standard tricks derived for Euclidean models, including the std trick, the residual trick, gradient clipping, etc. It seems likely we could significantly improve the relative performance of the constrained models with more time working on the training dynamics in higher dimensions. The MMD we use for evaluation uses a sum of weighted RBF kernels matching the generating distributions for the Gaussian mixture. This is well suited to the particular distribution we have generated, but our results do not change using a single RBF kernel or a Matern kernel. We have added a comment based on this discussion to the appendix with the Euclidean experiments.
> 3. We have added quantitative evaluations using the MMD for the protein case. Unfortunately we cannot compute the likelihood for the log-barrier method due to numerical instabilities. Running the likelihood evaluation for the logbarrier produces meaningless results.
>
> ## Theoretical Minor Points
>
> ### Proof of ODE
>
> Indeed there is a sentence missing at the end of the proposition. We have added the following sentence: Then for any $t \in [0,T]$, $X_t$ has the same distribution as $\bar{B}_t$.
>
> Since, whenever the two processes $(\bar{B}\_t)\_{t \in [0,T]}$ and $(X\_t)\_{t \in [0,T]}$ exist their densities satisfy the same equation, we need to check that these processes are indeed well defined everywhere.  This is true for the reflected Brownian motion using the classical theory of reflected SDE [1]. For the ODE there is some work involved. In particular, the bulk of the proof is to show that if we hit the boundary since then the score is smooth as a vector field on the \emph{boundary} of the manifold (which is itself a manifold) then the trajectory remains on the boundary However this is not possible, because in that case its time-reversal is also a flow on the boundary and therefore we do not recover the initial condition (which was not supported on the boundary).  The smoothness assumption is important here to ensure that once the process hits the boundary it then remains on the boundary.  Once we are ensured that the two processes remain in M (i.e. do not hit the boundary) we can conclude using the fact that the FP equations agree.
>
> This part of the proof is technical but has practical implications; if we start the process on the boundary then the ODE cannot leave the boundary while the SDE will. We will add a high level explanation of this proof.
>
> ### Reference to surface integral
>
> We now refer to [Proposition 2.43, 2] for a definition of the surface area. Note that a measure theoretic point of view is given in [3], where it is shown that the surface area corresponds to the Hausdorff measure of the boundary. Roughly speaking, the surface area corresponds to an equivalent of the Lebesgue measure defined w.r.t. the boundary.
>
> ### Convergence to the uniform distribution
>
> It is true that [4] only deals with convex domains. However, the exponential ergodicity of reflected Brownian motions is also known for smooth domains [5]. In [5], the authors show that the reflected Brownian motion is exponentially ergodic if and only if the domain is not a trap domain (see [5] for a definition) and remark that no smooth trap domain can be a trap domain. We have updated our manuscript with this remark.

---

> > ### Comment · Reviewer_x71F · 2023-06-08
> >
> > I thank the authors for their response. However, is the manuscript already updated? Many of the minor weaknesses are not addressed. Please go through them again. I would also suggest putting the Euclidean results to the main paper, as the difference in large dimensions (10) seems to be quite significant.
> >
> > I think there is no need for imaging experiments but the authors should put a dimension scaling remark in the main text. Could you please put the MMD results for the protein loop into perspective by adding the MMD of the uniform distribution on the manifold? MMD difference of factor 10 seems extremely large, as otherwise the difference between the methods was quite small.

---

> > > ### Author Response · Authors · 2023-06-08
> > >
> > > Apologies we uploaded the wrong manuscript! We have posted the version with the minor concerns addressed. We will address the new raised in this comment in our next revision.

---

> > > > ### Comment · Reviewer_x71F · 2023-06-08
> > > >
> > > > Thank you. Overall I am happy with the rest of the changes (modulo putting Euclidean in main text). Note that there is a missing reference in appendix G.

---

> ### Author Response · Authors · 2023-06-07
> **Response to Reviewer x71F (2 / 2)**
>
> ## Other Minor Points
>
> Finally we believe we have addressed all of the other minor issues except the point about Algorithm 2. We think it is helpful to completely spell out the reflected walk on manifolds, as that is one of the contributions of the work.
>
> ---
> [1] Lions, Sznitman (1984) – Stochastic differential equations with reflecting boundary conditions
>
> [2] Lee (2018) – Introduction to Riemannian manifolds
>
> [3] Evans, Gariepy (2015) – Measure theory and fine properties of functions
>
> [4] Loper (2020) – Uniform Ergodicity for Brownian Motion in a Bounded Convex Set
>
> [5] Burdzy, Chen, Marshall (2006) – Traps for reflected Brownian motion

---

### Review · Reviewer_KexQ · 2023-05-29

**Summary Of Contributions:**

This paper studies the design of diffusion models in constrained spaces. In particular, the diffusion models are built for Riemannian manifolds with a set of inequality constraints. Examples include polytopes and convex sets of Euclidean spaces. To facilitate a proper stochastic process in these constrained manifolds the authors propose two strategies: 1.) a logarithmic barrier where the inequality constrained is formulated using the geodesic distance to the boundary and 2.) a reflected diffusion model where the stochastic process is reflected back in the inward normal direction at the boundary. Using these two methods the paper proposes the construction of both a diffusion process and the time-reversed denoising process as well as sampling algorithms for each method. On the empirical side, the authors conduct experiments on toy synthetic domains, robotic arms under force and velocity constraints, and modeling protein loops with anchored endpoints. Finally, the experiments demonstrate that the reflected diffusion model slightly outperforms the log-barrier method.



**Audience:**

Yes

**Claims And Evidence:**

Yes

**Requested Changes:**

- A clearer distinction between contributions in this paper and ones borrowed from Lee and Vampala 2017.
- A more thorough discussion between the reflected diffusion model in this paper and the one in Lou and Ermon 2023. This I believe will be useful to the community at large.
- Inclusion of larger scale experiments on image data and comparisons to baseline diffusion models in this case.
- All results in Tables should have quantification of statistical uncertainty over multiple trials and not just the raw number. The current presentation is difficult to interpret because we don't know what the standard error is.



**Strengths And Weaknesses:**

Strengths:
- The paper addresses an important gap in the literature on designing diffusion models for constrained spaces. This problem is motivated by both theoretical considerations---i.e. De Bortoli et. al's approach cannot handle manifolds with constraints, and practical problems as domains such as robotics have rigid constraints that must be met.
- Conceptually, the main ideas in the paper are very clear. Both the log-barrier method and reflected diffusion model and their associated SDEs are natural fits for designing diffusion models in constrained spaces.
- The overall presentation of the material is clear and the writing is of high quality.

Weaknesses:
Overall the paper does not have a single major weakness but a series of minor ones that I list below.
- First, it seems both the main ideas---i.e. the log barrier method, and the reflected diffusion model have already been known to some degree in the literature. The technical hard part of the log-barrier method (designing the SDEs) can be found in Lee and Vampala 2017. While the reflected diffusion model appeared concurrently in Lou and Ermon 2023. So one must wonder about the amount of novelty here. I do admit that the authors use the ISM loss parametrization rather than DSM one---but its unclear if this is a meaningful enough differentiator. Can the authors comment on where the main differences between their contributions lie and the ones that already exist in the literature?
- Moreover, the reflected diffusion model in Lou and Ermon seems to have a much higher degree of execution. They scale it up to large-scale image datasets and show improved metrics over previous baselines. Since pixels within images that are represented in computers fall in the range of $[0, 255]$ and are often normalized to $[-1,1]^d$ we can think of them as a hypercube which is one the motivating examples used in this papers introduction. Consequently, I was surprised to see that there were no image experiments conducted in this paper. Also, one of the main claims in this paper is the scalability and flexibility of the proposed approach but the current experiments do not show this as the largest manifolds in the paper have $d=10$ (please correct me if I'm wrong here).
- I also suspect the scalability and flexibility of the log-barrier method. Specifically, the log-barrier method considered in this work requires the Hessian which results in a Hessian-manifold and this is only a specific type of barrier function that has a specific Riemannian metric $g(x) = A^TS^{-2}(x)A$. Thus the current paper does not seem that flexible in this regard. Furthermore, requiring the computation of a Hessian means that such an approach will be inherently harder to scale to manifolds of higher dimensionality. In fact, it is clear that this method is not scalable as evidenced by section 5.1 which requires the inversion of a $d \times d$ matrix which as the authors correctly remark requires $O(d^3)$ cost. Scaling this to the size of large-scale image datasets seems challenging.
- Lastly, for the log-barrier method warping the geometry of the problem seems like an undesirable tradeoff for generative modeling. The manifold itself is a rich object of study and by changing its structure we lose part of the problem definition so it seems not as elegant as the reflected diffusion model.

Question:
- The motivation at the beginning mentions that De Bortoli et. al and Huang et. al's approach cannot work with constrained manifolds. While I agree this is definitely the case for De Bortoli et. al who use an intrinsic perspective to parameterize the diffusion process which can escape the boundary I'm not entirely sure this problem exists with RDM as proposed in Huang et. al. Concretely, they use an extrinsic formulation and require a tangential projection operator so the diffusion automatically moves in the tangent space of the given manifold.  Perhaps I am wrong here but this would suggest that the process would not exit the manifold? I'm happy to be corrected here through a worked example.

Minor Comments:
- The formal approach to ’warping the geometry’ of" - Notice the first quote is formatted incorrectly in latex for this sentence found on page 4

---

> ### Author Response · Authors · 2023-06-07
> **Response to Reviewer KexQ (1 / 3)**
>
> We thank the reviewer for their thorough review and positive assessment of our work. Below we answer their concerns regarding the novelty of our work.
>
> ## Weakness 1: Novelty
> Regarding the log-barrier method. The forward process was indeed proposed in [1]. This does not correspond to the Brownian motion w.r.t. the Hessian metrics but to the Langevin dynamics (in Riemannian coordinates) targeting the uniform distribution. These facts are recalled in Appendix B. In that respect, we do not claim any theoretical novelty regarding the analysis of this process. Similarly, in the manifold setting, the time-reversal formula was derived in [2]. Hence, our log-barrier method is an application of the results of [1] and [2] to the case where the geometry is induced by the Hessian metric. Our contribution is mostly methodological and experimental; studying the practicality of log-barrier diffusion models.
>
> Regarding the reflected process, our manuscript is concurrent with the work of [3] and therefore we see our main contribution to be the introduction of reflected processes in the context of diffusion models, the same as [3].
>
> ## Weaknesses 2: Comparison with Lou and Ermon
> We now discuss a few key differences and similarities between our work and [3], when talking about the reflected SDE approach detailed in our work. First, in the hypercube setting, theoretically, our methodologies are identical, with the difference that in practice we use the ISM loss to train and [3] use an approximation version of the DSM loss. In our work, the focus was on settings relevant to scientific ML  (robotics, protein modeling) where the domains are typically not hypercubes or even Euclidean. [3] focus on the case where the constrained domain of interest is the hypercube (a convex subset of Euclidean space) with image applications, or where the domain can be easily projected into the hypercube, such as the simplex via a modified stick-breaking construction. It is not clear in [3] how such a projection would be achieved practically in the general case for convex sets, and so the application of this may be limited. We, therefore, see 2 settings the framework we put forward can handle but [3] cannot.
> - Bounded subsets of Euclidean space that are not convex or are convex but do not have a tractable projection into the hypercube, where we can no longer leverage the factorisation of the constraints on each dimension.
> - Any constrained set on manifolds, or combinations of unconstrained manifolds and constrained subsets. [3] is only formulated in terms of Euclidean space (where the metric is flat) whereas ours is formulated for more general manifolds. One can think of placing a distribution on the upper half of a sphere for example, or for example quantum tomography, which requires estimation of a distribution over the constrained space of SPD matrices, which would be very hard to appropriately model without leveraging the geometry of the manifold [8].

---

> > ### Comment · Reviewer_KexQ · 2023-06-12
> > **Re:Response**
> >
> > I thank the authors for their response to my novelty concerns. I am satisfied with the responses however I would like to see this discussion also be more prominent in the main paper. It is my belief that a reader would benefit from reading the specific settings that this paper can handle but Lou and Ermon 2023 cannot.

---

> ### Author Response · Authors · 2023-06-07
> **Response to Reviewer KexQ (2 / 3)**
>
> ## Weaknesses 2 and 3: Scaling when Constraints Factor
> Practically, the methodology of [3] has significant computational advantages on the spaces on which it works. These advantages can, however, be employed in our framework as well. The key factor is to exploit the decomposition of the constraints on each dimension, i.e. each constraint only depends on a single dimension of the state space. We intentionally did not do this in the current manuscript so the hypercube example would be more representative of the more complex geometries we investigate in our other experiments. This is the key to the scaling of the method in [3]. In the reflected setting this means that:
> - The reflection exponential map becomes analytic and can be done coordinatewise in $O(1)$ computation, for overall $O(d)$ computation. This is significantly faster than the generic case that has to be computed iteratively for each reflection.
> - The heat kernel becomes analytically tractable, as the hypercube is the product of $[0,1]^d$, and thus the heat kernel on the whole space is the product of the heat kernels on each [0,1], which is analytic (up to some expansion approximations). This takes $O(k)$ time per dimension where k is the number of approximation terms, for $O(kd)$ overall computation.
>
> Together these mean that it is possible to use (an approximation of)  the DSM loss, and take samples from the SDE evolved to time t directly without having to simulate the whole SDE. These methods can be employed exactly the same in our methods for the hypercube as in [3].
>
> The structure of the hypercube can also be exploited for the log-barrier method and would help with scaling significantly. For each coordinate of the space, we have a barrier at 0 and at 1. Denoting the distance to the closest boundary as $d(x_i) = \min(x_i, 1-x_i)$, a suitable barrier function is, therefore, $\phi_i(x) = \log(d(x_i))$, as this explodes at each boundary. Taking the sum of this over all dimensions gives us the full barrier function. Many of the quantities we need for the Hessian SDE become significantly more tractable.
> - The Hessian itself becomes a diagonal matrix with $1/d(x_i)^2$ as the diagonal elements.
> - The square root of the inverse metric matrix is therefore also a diagonal matrix, with $d(x_i)$ on the diagonal. Computing this is then $O(d)$ cost rather than $O(d^3)$, allowing for much better scaling.
> - The divergence of the inverse metric matrix need not be commuted by more expensive autograd options anymore, and is simply a vector with ith entry $d(x_i) d/dx(d(x_i))$, also $O(d)$ computation.
> - The exponential map with respect to this particular metric becomes analytic. We therefore no longer need to rely on a retraction to approximate the exponential map. This increases the accuracy of the SDE rollouts, and allows us to simulate the forward diffusion in many fewer steps than with the retraction.
>
> This will be detailed more in the appendix of the paper. We are currently implementing this approach, and applying it to the image base experiments asked for. We are unsure how much progress can be made on this in two weeks, but plan to test this on, in order, low dimension hypercubes (sanity check), MNIST, CIFAR 10, and see how far we can get before the end of the review period.

---

> > ### Comment · Reviewer_KexQ · 2023-06-12
> > **Re:Response to Scaling**
> >
> > I thank the authors for their response. I still believe image experiments are needed to demonstrate the scalability of the proposed approach. If they cannot be accomplished in the time frame I would ask the authors to kindly reword their claims and not mention scalability to higher dimensions as this has not been sufficiently well demonstrated.

---

> > > ### Author Response · Authors · 2023-06-15
> > > **Response to Reviewer KexQ re: Scaling**
> > >
> > > We have made progress the implementation required for image experiments during the rebuttal period, but do not have fully run experiments yet. Given it is the end of the rebuttal period now, we have tempered mentions of high-dimension scalability. It is our intention to finish these image experiments, and if possible include them at a later date. We have also added a significant discussion on the comparison of our work with that of Lou & Ermon.

---

> ### Author Response · Authors · 2023-06-07
> **Response to Reviewer KexQ (3 / 3)**
>
> ## Discussion of Huang et. al
>
> With regard to the comments regarding the framework of [4] being able to handle spaces with constraints:
> First, we highlight that the theoretical construction of [2] recovers the ones of [4] in the case where the manifold is isometrically embedded in a Euclidean space. Indeed some of the experiments in [2] are parametrised on a manifold embedded in Euclidean space. In that setting in practice the difference between [2] and [4] is that [2] uses analytic exponential maps, and [4] approximates these with the Euclidean exponential map followed by a closest point projection.  In fact, the theoretical treatment of [2], and more generally, the theory of stochastic processes on manifolds [5], heavily relies on the fact that any manifold can be isometrically embedded in a Euclidean space using the Nash embedding theorem.
>
> Next, we comment on the results of [4], and how it is not theoretically suitable to constrained domains, without any discretisation schemes. In [4] one of the implicit assumptions made by the authors is that the stochastic process exists for all time $t \in [0,T]$ for some constant $T>0$. This implies that the processes do not leave the manifold in finite-time. While this is ensured for stochastically complete manifolds [6], it is not always the case. For example, consider the hypercube $\mathcal{M} = [-1,1]^d$ in $\mathbb{R}^d$ and a Brownian motion $ (B_t)_{t \in [0,T]} $ initialized at 0. Under the framework of [4], we would use a standard Euclidean Brownian motion. However, since the Brownian motion leaves the hypercube with non-zero probability in finite time, it is clear that the process may not stay in the manifold for all time $t$. The stochastic process then does not exist for all times, and therefore the results of [4] break down and the model is no longer valid. This issue of finite time definition is referred to as the explosion time issue in the stochastic process literature [7].
>
> Looking more practically at the discretisation scheme, the projection operator used in [4] projects the tangent space of the ambient Euclidean space onto a subset that coincides with the tangent space of the embedded manifold. What this does not do is fix the fact that the exponential map we then push this tangent vector through to get to the next point in the discretisation of the SDE is not defined for regions of the tangent space that would place the next point outside the constraints. Dealing with this issue is in part the point of this work and of [3]. The final step of the discretisation in [4] is to approximate the exponential map on the manifold with the Euclidean exponential map and then a nearest point projection back onto the embedded manifold. This would return a point that has been sent out of the constraints on a manifold back inside them. What is not analysed in [4] is if this particular scheme is a proper discretisation of any particular SDE that would itself stay on the manifold.
>
> In summary:
> - The SDEs defined as the noising process in [4] are ill-defined for constrained domains.
> - The Euler-Maruyama discretisation (a valid discretisation) of these SDEs would still leave the manifold, as even with the projection operation one can sample tangent vectors that will send the next point outside the constraints.
> - The final step of the discretisation scheme in [4] of projecting points back inside the manifold does mean the discrete rollouts would live inside the constraints, but it is not clear that this is a proper discretisation of a corresponding SDE that would stay inside the constraints. In fact it can be shown that this discretisation is a valid discretisation for the reflected SDE, similar to [3]. However this departs from the theoretical framework proposed in [4].
> - [4] does not discuss nor perform experiments on constrained domains.
>
> ---
> [1] Lee, Vempala (2017) – Geodesic walks in polytopes
>
> [2] De Bortoli, Mathieu, Hutchinson, Thornton, Teh, Doucet (2022) – Riemannian score-based generative modelling
>
> [3] Lou, Ermon (2023) – Reflected Diffusion Models
>
> [4] Huang, Aghajohari, Bose, Panangaden, Courville – Riemannian diffusion models
>
> [5] Hsu (2002) – Stochastic analysis on manifolds
>
> [6] Grigor’yan (2011) – Stochastic completeness of Markov processes
>
> [7] Ikeda, Watanabe (2014) – Stochastic differential equations and diffusion processes
>
> [8] Jukens et al (2020) – A practical and efficient approach for Bayesian quantum state estimation

---

> > ### Comment · Reviewer_KexQ · 2023-06-12
> > **Re: Discussion of Huang et. al**
> >
> > I appreciate the authors taking the time to explain this point to me. I am now convinced by the statements made in the paper and I do not have any further contention on this point.

---

### Review · Reviewer_926N · 2023-06-01

**Summary Of Contributions:**

The paper considers the training of diffusion models on Riemanninian manifolds with inequality constraints. Diffusion models implicitly assume that the forward and time-reversed diffuison process used to perturb the data distribution are well-defined for all times. However, when the considered space upon which the diffusion evolves is a Riemannian manifold defined through inequaity constraints this is not necessarily the case. The authors tackle this problem by proposing to consider diffusion processes defined on the logarithmic barrier metric and with reflected Brownian motion induced by boundary constraints.

The first approach uses intuitions from barrier methods employed in optimisation and sampling, and considers a 'potential' that blows up at the boundary positions, which they introduce into the geometry of the considered space, that prevents the sampled process to hit the boundary. The resulting constrained diffusion is defined with respect to a metric that comes as the Hessian of the considered potential function.

The second approach introduces a reflected Brownian in the dynamics of the diffusion process that prevents the process to trespass the boundary of the domain.

The authors present numerical results of the two methods on applications which require sampling on constrained domains (like distributions symmetric positive definite matrices with trace constraints and conformational ensembles of proteins under position constraints). While the reflected method seems to work better on the experiments, and is claimed to scale better with system dimension, the evaluation of the likelihood for constrained models with the latter approach is not straightforward.

Overall, the paper is well written and provides a useful extension of diffusion modeling to bounded non-Eucledian domains. The proposed methods are supported with mathematical proofs in the supplement and numerical experiments, while the topic is relevant for the audience of the TMLR.





**Audience:**

Yes

**Broader Impact Concerns:**

There are no ethical implication directly related to the proposed methods.

**Claims And Evidence:**

Yes

**Requested Changes:**


- I think the authors should also cite [1] for the generalised score matching estimator on a generic domain $V$ that uses shortest distance to the boundary, which is effectively similar to the scaling function the authors considered here.

- For estimating the truncated loss the authors use a scaling function similar to the one used in [1] and a weighting function indicated as $\lambda_t$ in the text (function of time). Can the authors comment on the purpose of including the weighting function $\lambda_t$ that cannot be accommodated by the scaling function in Eq.(3) ? Also I think the expression of the loss should include also the scaling function to accurately reflect the implemented solutions.

- In page 3, after Eq. 2, the authors state that the $\nabla log p_t$ is unavailable in practice, but this is not true for the equation they mention above. For Ornstein-Uhlenbeck processes with additive noise the marginal density $p_t$ and thus also the Stein score have an analytically tractable solution. The process considered in [2] has time dependent diffusion and is therefore its marginal density is not analytically tractable. I think the authors probably should rewrite this sentence to refer to a more general case, or update the SDE to a multiplicative noise SDE.

- There are some recent approaches on constraining diffusions that employ score matching and the Doob-h transform to constrain diffusion processes [3,4] based on linearly solvable stochastic control [5]. Would such an approach defined for stochastic processes on manifolds provide an alternative solution for constraining the diffusion processes within the domain? I imagine this would result in altering the diffusion process to include an extra drift term that would be similar to a potential that blows up at the boundary. But as I understand this would not result in a similar formulation as the logarithmic barrier approach proposed here.

- Figure 9 needs a better explanation in the caption on what is depicted. Also for completeness I would consider necessary to provide in the supplement the same plot with the omitted samples.


**Minor:**

- page 3, middle: the sentence “(iv) A score matching loss, in this paper we will focus on the implicit score matching loss.” reads strange. I think it requires a full stop or a  dash after the first loss.

- In Algorithm 1: for completenes explain the parameters T,N X, b that the algorithm requires as input. As far as I understand $N$ is the number of discretisation steps, and $T$ the total simulation time, b the drift of the forward process, and $X$ the initial condition (?). Also please consider to use a different letter to indicate the drift $b$ since in the previous page $b$ was a parameter used to define the inequality constraints.
- Page 5 last paragraph: I think the standard metric with the previous notation on the unconstrained domain $\mathcal{N}$ is denoted by $\mathfrak{h}$ not $\mathfrak{g}$.

- The hyperlinks of the pdf files do not seem to work and given the length of the paper and the supplement it is difficult to read.
------
**References:**

[1] Liu, S., T. Kanamori, and D. J. Williams. "Estimating density models with truncation boundaries." Journal of Machine Learning Research (2022)

[2] Song, Yang, et al. "Maximum likelihood training of score-based diffusion models." Advances in Neural Information Processing Systems 34 (2021)

[3] Zhang, Benjamin, Tuhin Sahai, and Youssef Marzouk. "Sampling via Controlled Stochastic Dynamical Systems." I (Still) Can't Believe It's Not Better! NeurIPS 2021 Workshop (2021)

[4] Maoutsa, D., and Opper, M. "Deterministic particle flows for constraining stochastic nonlinear systems." arXiv preprint arXiv:2112.05735 (2021)

[5] Kappen, Hilbert J. "Linear theory for control of nonlinear stochastic systems." Physical review letters 95.20 (2005).

**Strengths And Weaknesses:**

**Strengths:**

- The authors propose two methods for diffusion modeling relevant for problems defined on Riemannian manifolds with boundary conditions.
- Extensive numerical experiements.
- The paper is mathematically sound, with detailed derivations in the supplement.




**Weaknesses:**

- The log barrier method does not seem to work so well for constraining the diffusion in the numerical experiments (especially the ones in the supplement). Can the authors comment on that? What is the reason behind this and how could it potentially mitigated?
- The scaling function used for estimating the score on the truncated domain has been actually introduced before, see the comments below.

---

> ### Author Response · Authors · 2023-06-07
> **Response to Reviewer 926N**
>
> We thank the reviewer for their thorough comments. First we want to note that we actually have solved the likelihood evaluation for the reflected diffusion and it is actually already in the appendix; as Reviewer x71F notes the comment in the summary is actually an artifact of an older manuscript. Now addressing the weaknesses in order.
>
> ## Major Concerns:
> 1. We have added [1] as a citation for the weighting; we had not seen this reference and appreciate it being pointed out.
> 2. There are two components we are weighting. First we are directly rescaling the score output by the network to scale to zero at the boundary. This is a position dependent/time independent weighting function applied before the loss. Second we are doing the standard diffusion reweighting of the loss according to the time the loss is computed at. This reweighting of the loss is meant to ensure earlier stages in time are learned better on the premise that if you fail to learn the score at early stages you can have arbitrarily poor samples irrespective of learning at later stages. We need to modify this weight function to work in our setting; it is unclear why the default weighting works so poorly. Given these two weighting functions affect two different objects, we cannot combine these two weighting functions.
> 3. We give a more thorough discussion of this point below.
> 4. We have actually found a way to visualize the full samples in a way that is clear. Unfortunately the author who worked on these visualizations is currently unavailable. We will update the visualizations as soon as possible.
> ## Minor Concerns
> 1. we have updated the sentence
> 2. we have added the descriptions to the inputs to all algorithms
> 3. this is correct, we’ve updated the metric to be \mathfrak{h}
> 4. we believe this is fixed.
>
> ## Discussion of the Third Major Weakness:
> First, we emphasize that $\nabla \log p_t$ is intractable. This is because, while $\nabla \log p_{t|0}$ i.e. the Stein of the distribution conditioned on its initial point is tractable (since $p_{t|0}$ is Gaussian with mean and covariance that can be computed), the general score cannot be computed since $p_0$ is not known. If we had access to $\nabla \log p_t$ then we could bypass the learning of this quantity.
> Regarding the reference to [3,4]. The control point of view on diffusion models has been studied in several works [1,2]. In particular, [Equation (4), 3] is very similar to the loss of diffusion models except that we do not integrate with respect to the control process. This is a key difference which makes diffusion models easier to train.
>
> If we understand correctly what is suggested by the reviewer, we could modify the forward process to incorporate the constraints of the model. Then, the time-reversal could still be computed (approximating the score using the ISM loss). One drawback of this method is that 1) if the drift does not blow up near the boundary then it is possible for the forward dynamics to cross the boundary and therefore violate the constraints, 2) unless we choose the drift in an appropriate manner it is not clear what the invariant distribution of the process is. The invariant distribution is necessary  to start the time-reversal. In addition, this invariant distribution needs to be as simple as possible, in order for the sampling to be efficient. Log-barrier and reflected processes are so far the only two methods we are aware of which provably converge to the uniform distribution on bounded manifolds. Finally, we emphasize that links with the Doob h-transform have been considered in the diffusion model literature [5,6] but in the context of Diffusion Schrodinger Bridge.
>
> ---
> [1] Berner, Richter, Ullrich (2022) – An optimal control perspective on diffusion-based generative modeling
>
> [2] Zhang, Katsoulakis (2023) – A mean-field games laboratory for generative modeling
>
> [3] Zhang, Benjamin, Tuhin Sahai, and Youssef Marzouk (2021) -Sampling via Controlled Stochastic Dynamical Systems
>
> [4] Maoutsa, and Opper (2021) - Deterministic particle flows for constraining stochastic nonlinear systems
>
> [5] De Bortoli, Thornton, Heng, Doucet (2021) – Diffusion Schrödinger bridge with applications to score-based generative modeling
>
> [6] Somnath, Pariset, Hsieh, Martinez, Krause, Bunne (2023) – Aligned Diffusion Schrodinger Bridges

---

> ### Author Response · Authors · 2023-06-15
> **Follow-up Response to Reviewer 926N on Fig. 9**
>
> To address the concern raised regarding Figure 9, we have updated its caption to convey a better understanding of what it depicts and have updated it with the improved visualizations that we referred to in our previous comment, allowing us to visualize all samples from the true and learnt distribution in a clear manner. It should now be possible to intuitively see what plots 23 and 24 show in more detail - that is that the log-barrier method produces a relatively small (<3%) number of samples that are significantly closer to the boundary than the data distribution, resulting in uncharacteristically large ellipsoids. We note that this is not an issue with the reflected Brownian motion-based model.

---

### Decision · Action_Editors · 2023-07-10

**Recommendation:** Accept as is

**Comment:**

The conclusion of the rebuttal process is that the paper is well written and contains sufficient evidence to back up their main claims. The authors toned down claims of scalability of the proposed method because of the lack of evidence that supports this claim. There are several domains of applications for diffusion models on constrained domains, and the topic is also of interest from a theory point of view. Therefore we expect this manuscript to be of interest to the audience of TMLR, and create opportunities for interesting follow-up work. Although the novelty of the proposed method is perhaps limited compared to prior/concurrent work, the manuscript nicely exposes the similarities and differences with this related work. Taking all this into consideration, the recommendation is to accept the manuscript as is.

**Audience:**

All reviewers agree that the topic of the paper is interesting both from a theoretical and practical point of view and that it will be of interest to the community who works on diffusion models. Although the topic of the paper is very technical, the quality of writing is good, increasing the chances of this manuscript being picked up with the community for follow up work.

**Claims And Evidence:**

All reviewers agree that the authors have backed up their claims with clear evidence. The only claim that was lacking evidence was around the scalability of the method. These claims have been toned down in the manuscript after the rebuttal.